# DETECTING MODULARITY IN DEEP NEURAL NETWORKS

## ABSTRACT

A neural network is modular to the extent that parts of its computational graph (i.e. structure) can be represented as performing some comprehensible subtask relevant to the overall task (i.e. functionality). Are modern deep neural networks modular? How can this be quantified? In this paper, we consider the problem of assessing the modularity exhibited by a partitioning of a network's neurons. We propose two proxies for this: *importance*, which reflects how crucial sets of neurons are to network performance; and *coherence*, which reflects how consistently their neurons associate with features of the inputs. To measure these proxies, we develop a set of statistical methods based on techniques conventionally used to interpret individual neurons. We apply the proxies to partitionings generated by spectrally clustering a graph representation of the network's neurons with edges determined either by network weights or correlations of activations. We show that these partitionings, even ones based only on weights (i.e. strictly from non-runtime analysis), reveal groups of neurons that are important and coherent. These results suggest that graph-based partitioning can reveal modularity and help us understand how deep neural networks function.

## 1 INTRODUCTION

Modularity is a common property of complex systems, both natural and artificial (Clune et al., 2013; Baldwin & Clark, 2000; Booch et al., 2007). It carries advantages including intelligibility and adaptivity. In particular, modularity implies that the structure of a system conveys information about its functionality. This leads us to the following definition: a neural network is modular to the extent that parts of its computational graph (structure) can be represented as performing some comprehensible subtask relevant to the overall task (functionality). We call these parts *modules*.

There exists a body of research for developing more modular networks which either have distinct architectural building blocks (Alet et al., 2018; Parascandolo et al., 2018; Goyal et al., 2019) or are trained in a way that promotes modularity via regularization or parameter isolation (Kirsch et al., 2018; De Lange et al., 2019; Filan et al., 2021). Yet in machine learning, it is more common to encounter networks whose architecture and training are not guided by considerations of modularity. For example, in computer vision, networks are generally trained end-to-end with all of the filters in one layer connected to all filters in the next. Do such networks nonetheless develop modularity? There is some evidence for this. For example, by methodical manual investigation, Cammarata et al. (2020) discovered modular subnetworks which perform human-explainable subtasks such as car detection via neurons which detect different car parts. More scalably detecting modularity in networks would help us to extend our understanding of their learning dynamics and to expand our interpretability toolbox by suggesting an additional level of abstraction beyond single-neuron methods.

In this paper we systematically analyze the extent to which networks which are not explicitly trained to be modular exhibit it nonetheless. First, this requires a method for breaking down a network's computational graph into different proposals for modules. For this, we use spectral clustering on a graph representation of the network, using an extension of the methods of (Filan et al., 2021). Second, this requires a scalable method for approximating the degree to which a partitioning exhibits modularity. We do this by applying interpretability tools to clusters of neurons as a way of quantifying proxies for modularity.

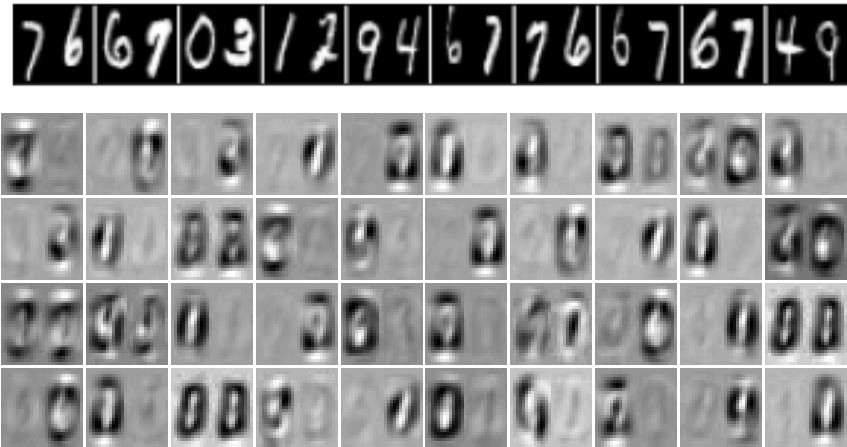

Figure 1: **An illustrative example of coherence.** Randomly selected examples of class "3" from our "halves-MNIST" dataset (top) and visualizations of neuron clusters in the first layer of networks trained to output the modular sum of digits in the images (bottom). We trained multilayer perceptrons on a version of the MNIST dataset (LeCun et al., 1998) in which the images were two half-width digits side-by-side and the labels were their sum modulo 10. We clustered their first-layer neurons using approaches presented in Section 3.1. We then used a correlation-based method from Watanabe (2019) to create visualizations of the clusters. Almost all show selectivity to one half of the input. In Appendix A.4, we detail our approach and compare to visualizations of random sets of units.

**Operationalization of modularity:** Following our notion of modularity, to say that a subset is a module, one must show that it performs some human-comprehensible subtask relevant to the overall task. Scaling this, however, would be difficult, as it would require a human in the loop. How could measuring modularity be automated? Consider an idealized prototype of a highly modular network that has subsets of neurons, each performing subtasks in which (1) each subtask is necessary for high performance on the overall task, (2) each subtask is implemented by a single subset only, and (3) each subset executes only a single subtask. The combination of (1) and (2) suggests that the removal of one of the subsets from the network would harm performance because the network lacks the implementation of a necessary subtask. We say that such a subset is *important*. Next, given that neurons are frequently understood as feature detectors, (3) suggests that the neurons in a module should tend to be strongly activated by inputs which contain features relevant to the module's subtask. We say that such a subset is *coherent*. Figure 1 provides an illustrative example of coherence in networks that are trained on a task that lends itself to parallel processing via subtasks. Measuring importance and coherence will give us a sense of the degree to which networks approximate these prototypical conditions, despite not perfectly satisfying them.

**Results and contributions:** To measure these proxies, we utilize interpretability methods that have been conventionally used for single-neuron analysis to study these partitions. We find that the partitions have groups of neurons that are disproportionately likely to be important compared to random groups of neurons. We also find that the groups of neurons in the partitionings are reliably more coherent than random ones, though only with respect to features other than class label.

By showing that our partitioning methods are able to reveal modularity, these results suggest that they can be used to understand deep networks. Our key contributions are threefold:

1. Introducing two proxies, importance and coherence, to assess whether a partitioning of a network exhibits modularity and identify which subsets of neurons are the most responsible.

2. Quantifying these proxies by applying single-neuron interpretability methods on subsets of neurons in an automated fashion

3. Applying our methods on the partitions produced by spectral clustering on a range of neural networks and finding evidence of modularity captured by these partitions.

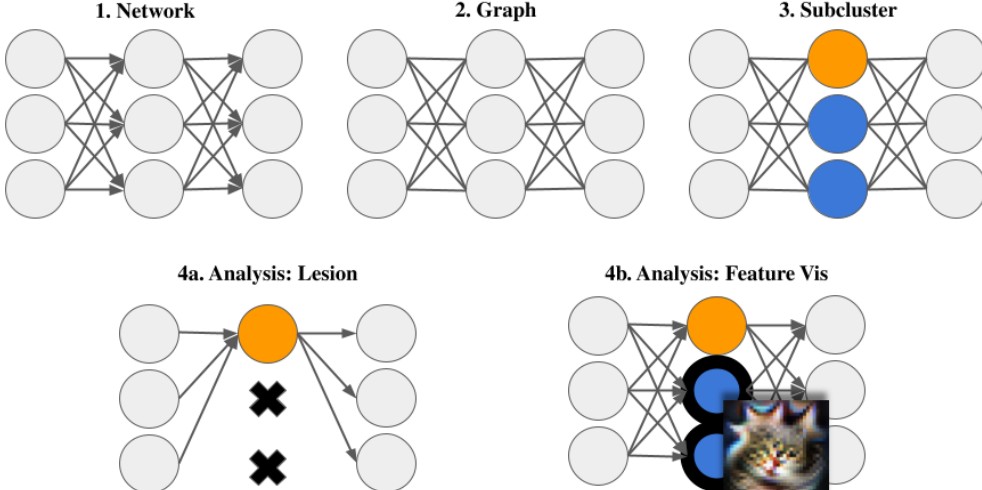

Figure 2: **Our procedural pipeline.** The first three steps generate a partitioning of the network into "subclusters" which we analyze using (4a) lesion and (4b) feature visualization methods to measure importance and coherence compared to random subclusters. Finally, (not shown in the pipeline), we aggregate results to produce Fisher-Bates $p$ values and effect measures. These final steps are shown in Figure 4.

## 2 RELATED WORK

The work most closely related to ours is Filan et al. (2021) who also use spectral clustering to establish that deep networks are often clusterable and investigates what factors influence clusterability. They also introduce a method for regularization for modularity among clusters of neurons. We extend their work by bridging graphical clusterability and functional modularity. This line of work inherits insights from network science involving clustering in general (Girvan & Newman, 2002; Newman & Girvan, 2004), and spectral clustering (Shi & Malik, 2000; von Luxburg, 2007) in particular.

In our experiments we combine clustering with interpretability tools to measure importance and coherence. This highlights the usefulness of combining clustering with interpretability methods for more rigorously understanding networks. We use feature visualization (Olah et al., 2017; Watanabe, 2019) and lesions (Zhou et al., 2018), but in a similar way, other interpretability techniques including analysis of selectivity (Morcos et al., 2018; Madan et al., 2020), network "dissection" (Bau et al., 2017; Mu & Andreas, 2020), earth-mover distance (Testolin et al., 2020), or intersection information (Panzeri et al., 2017) could also be combined with clustering-based partitionings under a similar framework. Relatedly, Cammarata et al. (2020) demonstrate that feature visualization and analysis of weights can be used to identify groups of neurons whose functionality is human-interpretable.

This work adds to a body of research focused on modularity and compositionality in neural systems either at the neuron level (You et al., 2020; Mu & Andreas, 2020; Voss et al., 2021) or at the subnetwork level (Lake et al., 2015; 2017; Csordás et al., 2021). There also exist techniques for developing more modular networks which either have an explicitly modular architecture (Alet et al., 2018; Parascandolo et al., 2018; Goyal et al., 2019) or are trained in a way that promotes modularity via regularization or parameter isolation (Kirsch et al., 2018; De Lange et al., 2019).

## 3 METHODS

To evaluate modularity, our procedural pipeline is as follows. (1) We begin with a trained neural network; (2) construct a graph from it, treating each neuron as a node; (3) perform spectral clustering on the graph to obtain a partitioning or "clustering" of neurons which we then further divide by layer to obtain a "subclustering"; (4) analyze the subclusters according to our modularity proxies operationalized by lesioning neurons or feature visualization and comparing them to random subclusters;

and (5) aggregate results across the network to obtain a final $p$ value and effect measure. This pipeline is outlined in Figure 2 and Figure 4, and each step is explained in detail below.

### 3.1 Generating Partitions with Spectral Clustering

To partition a network into clusters, we use an approach based on Filan et al. (2021) which consisted of two steps: *"graphification"* - transforming the network into an undirected, edge-weighted graph; and *clustering* - obtaining a partitioning via spectral clustering.

**Graphification:** To perform spectral clustering, a network must be represented as an undirected graph with non-negative edges. For MLPs (multilayer perceptrons), each graph node corresponds to a neuron in the network including input and output neurons. For CNNs (convolutional neural networks), a node corresponds to a single channel (which we also refer to as a "neuron") in a convolutional layer.[1] For CNNs, we do not use input, output, or fully-connected layers for clustering.

For graphification, we test two ways of assigning adjacency edges between neurons: with weights and with correlations. For weight-based clustering with dense layers, if two neurons have a weight connecting them in the network, their corresponding vertices are connected by an edge with weight equal to the absolute value of the network's weight between the neurons. For convolutional channels, we connect them by an edge with weight equal to the $L_1$ norm for the corresponding 2D kernel slice. If layers are connected but with a batch-normalization layer in between, we mimic the scaling performed by the batch norm operation by multiplying weights by $\gamma/(\sigma + \varepsilon)$ where $\gamma$ is the scaling factor, $\sigma$ is the moving standard deviation, and $\varepsilon$ is a small constant. Notably, this method of constructing the graphs requires no dataset or runtime analysis of the network.

While graphification via weights only results in connections between neurons in adjacent layers, doing so with correlations creates more dense graphs. With this method, we connect the nodes for two neurons with their squared Spearman correlation across a validation set. Spearman correlation gives the Pearson (linear) correlation between ranks and reflects how well two sets of data can be related by a monotonically increasing function.[2] Rather than using neurons' post-ReLU outputs to calculate these correlations, we use their pre-ReLU activations with the goal of extracting richer data from neurons.[3] Again, for convolutional channels, we take the $L_1$ norm of activations before calculating Spearman correlations. The fact that we take absolute valued weights or squared correlations to construct edges between neurons in graphification means that our analysis does not discriminate between positive and negative associations between neurons.

In addition to graphification via weights versus activations, we also test two scopes in which to perform clustering: global and local. For global clustering, we cluster on one graph for the network as a whole. For local clustering, we produce a partitioning for each layer $l$ individually by clustering on the graph of connections between $l$ and the layers adjacent to $l$. Ultimately, we run 4 experiments on each network by clustering {weights, activations} $\times$ {globally, locally}.

**Spectral (sub)clustering:** We perform normalized spectral clustering (Shi & Malik, 2000) on the resulting graphs to obtain a partition of the neurons into clusters. For all experiments, we set the number of clusters to $k = 16$. In Appendix A.6, we reproduce a subset of experiments with $k \in \{8, 12\}$, showing that results are robust to alternate choices of $k$. Refer to appendix A.1 for a complete description of the spectral clustering algorithm.

Layers at different depths of a network tend to develop different representations. To control for these differences, we study the neurons in clusters separately per layer. Therefore, for global clustering in which clusters of neurons span more than one layer, we analyze clusters one layer at a time. We call these sets of neurons within the same cluster and layer *subclusters*. To ensure comparability between these clusterings when performing local clustering, we set the number of clusters per layer to be the same as the number produced in that layer with global clustering. In our experiments, we compare

---

[1]This is subtly different from treating each activational unit as a neuron in networks such as ResNets with skip connections. If a unit is used as inputs to multiple layers, we consider these inputs to be separate neurons.

[2]We use Spearman rather than Pearson correlation because networks are nonlinear, and there is no particular reason to expect associations between arbitrary neurons to be linear.

[3]Although all negative values are mapped to zero by a ReLU, we expect the degree of negativity to carry information about the possible presence (or lack thereof) of features which the neuron was meant to detect, especially for networks trained with dropout.

these subclusters to other sets of random units of the same size in the same layer. When discussing these experiments, we refer to subclusters from the clustering algorithm as "true subclusters" and sets of random neurons as "random subclusters." Random subclusters form the natural control condition to test whether the specific partitioning of neurons exhibits importance or coherence compared to alternative partitions, while taking account of layer and size.

## 3.2 ANALYSIS PIPELINE

**Summary:** In our experiments, we measure the degree to which subclusters identified via spectral clustering are more important and coherent than random sets of neurons of the same size and layer. Operationalizations are given in subsections 4.1 and 4.2, but in brief, the measure for importance of a subcluster quantifies the performance reduction from dropping out or "lesioning" the neurons in that subcluster, and the measures for coherence quantify the degree to which the neurons in a subcluster are mutually associated with some feature of an input. For each subcluster greater than 1 neuron in size and which did not include every neuron in the layer, we calculate a measure of importance or coherence and compare it to that of 19 random subclusters for non-ImageNet networks or 9 random subclusters for ImageNet ones. These experiments and measures are discussed in detail in section 4. We present two measures of how true and random subclusters compare under these proxies. First, we test whether true subclusters are disproportionately often more important or coherent than random subclusters, and report a $p$ value for this test, which we call a Fisher-Bates $p$ value. This value is our primary focus. However, for additional resolution, we also calculate an effect measure used to assess the importance and coherence of the 'average' subcluster. Importantly, the Fisher-Bates $p$ value and effect measure are based on different statistics and measure different things.

**Fisher-Bates $p$ values:** We wish to test whether spectral clustering-based partitioning methods find more modular subsets of neurons than simply choosing random subsets. To do this, for each subcluster measurement we take the percentile of each true subcluster relative to the distribution of measurements of random subclusters.[4] Next, we use the Fisher method to test whether the subclusters in a single network are disproportionately more modular than random neurons. To do so, we first center the subcluster percentiles around 0.5, which under the null hypothesis would give a granular approximation of the uniform distribution. We then combine the centered percentiles $\{p_1, \ldots, p_n\}$ into the test statistic $-2 \sum_{i=1}^{n} \log p_i$, which under the null hypothesis is drawn from a chi-squared distribution with $2n$ degrees of freedom. We can then produce a $p$ value for each network, testing whether this statistic is higher than the null hypothesis would produce, which would mean that there were more low percentiles than if they were distributed uniformly.[5] This procedure is illustrated in Figure 4.

For all non-ImageNet networks, we train and analyze 5 different versions of each which requires a second aggregation step. In each condition, we take the mean of the $p$ values for each network and correct it using the corresponding quantile of a Bates($n = 5$) distribution[6] which gives the distribution of the mean of five independent uniformly-distributed random variables.[7]

This produces a $p$ value for every network architecture, partitioning method, and modularity metric, which we call Fisher-Bates $p$ values.[8] We next correct for multiple testing using the Benjamini Hochberg method (Benjamini & Hochberg, 1995), controlling the false discovery rate at the $\alpha = 0.05$ level. See Appendix A.7 for details. Figure 3 shows which values are significant after this correction.

In summary, (1) we first aggregate each network's subcluster percentiles using the Fisher method, (2) then we aggregate these new $p$ values among identically-configured replicates of the same experiment using the Bates method, and finally (3) we correct for multiple comparisons using the Benjamini-Hochberg procedure.

---

[4] For metrics where high values indicate modularity, we take the percentile of the negative metric, so that low percentiles consistently indicate modularity.

[5] The fact that we coarsely measure percentiles and then center them makes this test conservative, because our statistic is more sensitive to low percentiles than high percentiles.

[6] The Bates($n$) quantile function is $F_n(x) = \frac{1}{n!} \sum_{k=0}^{\lfloor nx \rfloor} (-1)^k \binom{n}{k} (nx - k)^{n-1}$ (Marengo et al., 2017).

[7] We use the Bates method for aggregation here instead of using the Fisher method again because in this case, all five $p$ values are from identically configured experiments, and the Bates test is less sensitive to low outliers.

[8] For simplicity, we use this name even for ImageNet networks, where we do not have multiple networks to apply Bates aggregation to.

**Obtaining effect measures:** In addition to $p$ values, we also calculate effect measures which give a sense of how different our results for true and random subclusters are *on average*. We calculate them by taking the mean of $2x/(x + \mu)$ where $x$ is a true subcluster measure of importance/coherence and $\mu$ is the mean over that of random subclusters.[9] We do this as opposed to simply taking $x/\mu$ in order to avoid divisions by zero. This results in effect measures in the interval [0,2], and which side of 1 they are on indicates whether the true subclusters are more important/coherent than random ones. For ease of interpretation, note that if $2x/(x + \mu) = 1 + y$, then $x/\mu \approx 1 + 2y$ for $y \ll 1$, so an effect measure of 1.05 would mean that the measure of a true subcluster was $\approx 10\%$ higher than the expected measure of a random subcluster. Together with these effect measures, we also report their standard errors.

**Differences between Fisher-Bates $p$ value and effect measure results:** In some of our experiments, the Fisher-Bates $p$ values and effect measures seem to "disagree" with one suggesting that modularity was detected and the other suggesting it was not. This potential for disagreement is due to the fact that the Fisher-Bates $p$ values are based on the percentile of subcluster measurements relative to the distribution of those of random subclusters, while the effect measures compare the value of true subcluster measurements to the mean value of random subcluster measurements. When the Fisher-Bates $p$ value seems to indicate modularity but the effect measure does not, this means that on the relevant metric, there are more subclusters than would be expected under the null hypothesis whose metric value is higher than that of random subclusters, but the average subcluster has a metric value similar to or less than an average random subcluster. In other words, the partitioning method has detected some subclusters that are modular, but the average subcluster found was not modular. We consider this a positive result, indicating that our partitioning method is detecting some modularity.

## 4 EXPERIMENTS

To show the applicability of our methods at different scales, we experiment with a range of networks. For small-scale experiments, we train MLPs with 4 hidden layers of 256 neurons each and small convolutional networks with 3 layers each of 64 neurons followed by a dense layer of 128 neurons trained on the MNIST (LeCun et al., 1998) dataset. At a mid scale, we train VGG-style CNNs containing 13 convolutional layers using the architectures of Simonyan & Zisserman (2014) trained on CIFAR-10 (Krizhevsky & Hinton, 2009) using the procedure of Liu & Deng (2015), which includes weight decay and dropout for regularization. Finally, for the ImageNet (Krizhevsky & Hinton, 2009) scale, we analyze pretrained ResNet18 (He et al., 2016), VGG-16, and VGG-19 (Simonyan & Zisserman, 2014) models. Further details including hyperparameters and test performances are in Appendix A.2.

### 4.1 LESION EXPERIMENTS

One approach that has been used for understanding both biological (Gazzaniga & Ivry, 2013) and artificial (Zhou et al., 2018; Casper et al., 2020) neural systems involves disrupting neurons during inference. We experiment with "lesion" tests in which we analyze network performance on the test set when a subcluster is dropped out. We then analyze the damage to the network's performance. First, we measure importance by taking the drop in accuracy. Specifically, let $\theta$ be the parameter vector of the neural network, $c$ be a set of neurons, $\mathcal{M}(\theta, c)$ be a masked version of $\theta$ where weights into or out of nodes in $c$ have been set to 0, and $\text{Acc}(\vartheta, \mathcal{D})$ be the accuracy of the network parameterized by $\vartheta$ on dataset $\mathcal{D}$. Then, our measure for importance is $\text{Acc}(\theta, \texttt{test}) - \text{Acc}(\mathcal{M}(\theta, c), \texttt{test})$, where $\texttt{test}$ is a test dataset that was not used to construct the activation-based partitionings.

Second, we measure the coherence in a subcluster with respect to class by taking the range of class-specific accuracy drops. Specifically, let $\texttt{test}_i$ be the subset of the test set with label $i$, and let $\Delta(\theta, c, i) := \text{Acc}(\theta, \texttt{test}_i) - \text{Acc}(\mathcal{M}(\theta, c), \texttt{test}_i)$ be the drop in accuracy for examples with label $i$ from lesioning $c$. Then, this measure of coherence is the range $(\max_i \Delta(\theta, c, i)) - (\min_i \Delta(\theta, c, i))$, of accuracy drops over classes. We use this to detect whether clusters are more crucial for some classes over others, which would suggest that they coherently act to correctly label those classes.

We use the analysis pipeline from Section 3.2 to test for importance and coherence using these overall accuracy differences and class-wise ranges. Low Fisher-Bates $p$ values suggest the detection of

---

[9]If this value is ever less than 0 for a subcluster, we conservatively replace it with 0.

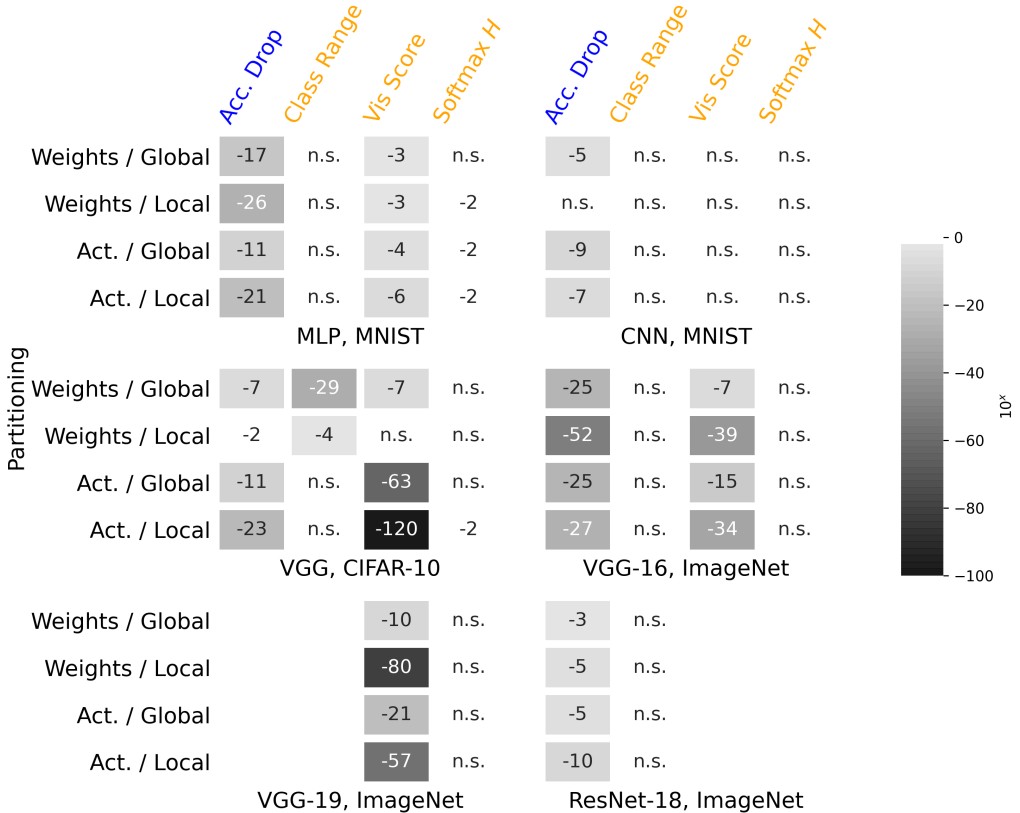

Figure 3: Fisher-Bates $p$ values for (1) lesion-based experiments measuring importance via overall accuracy drops (Acc. Drop) and coherence via the range of class-wise accuracy drops (Class Range); and (2) feature visualization-based experiments measuring coherence via the optimization score of feature visualizations (Vis Score) and the entropy of network outputs (Softmax $H$). Each subfigure corresponds to a type of network, where each row represents a partitioning method and each column a modularity proxy measurement. The Fisher-Bates $p$ values are shown on a log scale rounded to nearest integer. If a $p$ value is significant at an $\alpha = 0.05$ level after Benjamini Hochberg correction ($p \leq p_{\text{crit}} = 0.025 \approx 10^{-1.6}$), then its background is shaded. Otherwise, "n.s." (not significant) is reported. Results are calculated as described in Section 3.2.

an unexpectedly high number of important/coherent subclusters. Meanwhile, effect measures $> 1$ indicate more importance/coherence among true subclusters on average compared to random ones. Figure 3 shows Fisher-Bates $p$ values, and Table 1 shows effect measure data. Results are summarized in section 4.3.

## 4.2 FEATURE VISUALIZATION EXPERIMENTS

To further analyze coherence, we leverage another set of interpretability techniques based on feature visualization. We use gradient-based optimization to create an input image which maximizes the $L_1$ norm of the pre-ReLU activations of the neurons in a subcluster. Letting the parameter vector be $\theta$ and the subcluster be $c$, we write $\text{Act}(x, \theta, c)$ for the vector of pre-ReLU activations of neurons in $c$ in network $\theta$ on input $x$, and denote this optimized input image as $x(\theta, c)$, which approximately maximizes $\|\text{Act}(x, \theta, c)\|_1$. The key insight is that properties of these visualizations $x(\theta, c)$ can suggest what roles the subclusters play in the network. Implementation details are in Appendix A.5, and Figure 7 shows example visualizations.

We use two techniques to analyze coherence using these visualizations of subclusters. First, we analyzed the value of the maximization objective for each image we produced, $\|\text{Act}(x(\theta, c), \theta, c)\|_1$, which we call the "score" of the visualization. This gives one notion of how coherent a subcluster may

Table 1: Effect measures for (1) lesion-based experiments measuring importance via overall accuracy drops (Acc. Drop) and coherence via the class-wise range of accuracy drops (Class Range); and (2) Feature visualization-based experiments in networks measuring coherence via the optimization score (Vis Score) and the entropy of network outputs (Softmax $H$). Each row corresponds to a network paired with a partitioning method. Results are calculated as explained in Section 3.2. For accuracy drop, class-wise range and visualization score experiments, an effect measure $> 1$ corresponds to more importance/coherence among true subclusters than random ones, while one of $< 1$ does for softmax entropy experiments. Entries where the effect measure is more than two standard errors away from 1 in the direction of modularity are **bolded**.

| Network | Partitioning | Lesion | | Feature Visualization | |
| | | Acc. Drop High→Imp. | Class Range High→Coh. | Vis Score High→Coh. | Softmax H Low→Coh. |
|---|---|---|---|---|---|
| MLP, MNIST | Weight/Global | $\mathbf{1.123 \pm 0.058}$ | $0.701 \pm 0.036$ | $1.003 \pm 0.004$ | $1.105 \pm 0.021$ |
| | Weight/Local | $1.061 \pm 0.048$ | $0.676 \pm 0.029$ | $\mathbf{1.024 \pm 0.004}$ | $\mathbf{0.931 \pm 0.016}$ |
| | Act./Global | $0.883 \pm 0.038$ | $0.646 \pm 0.024$ | $\mathbf{1.02 \pm 0.003}$ | $0.997 \pm 0.013$ |
| | Act./Local | $0.929 \pm 0.040$ | $0.687 \pm 0.025$ | $\mathbf{1.026 \pm 0.003}$ | $1.011 \pm 0.012$ |
| CNN, MNIST | Weight/Global | $0.837 \pm 0.048$ | $0.527 \pm 0.031$ | $0.998 \pm 0.004$ | $1.026 \pm 0.008$ |
| | Weight/Local | $0.814 \pm 0.046$ | $0.635 \pm 0.033$ | $1.004 \pm 0.003$ | $1.007 \pm 0.007$ |
| | Act./Global | $1.078 \pm 0.061$ | $0.543 \pm 0.039$ | $0.933 \pm 0.006$ | $1.025 \pm 0.011$ |
| | Act./Local | $0.939 \pm 0.060$ | $0.625 \pm 0.043$ | $0.925 \pm 0.005$ | $\mathbf{0.970 \pm 0.010}$ |
| VGG, CIFAR-10 | Weight/Global | $0.682 \pm 0.066$ | $0.407 \pm 0.042$ | $0.871 \pm 0.011$ | $1.124 \pm 0.012$ |
| | Weight/Local | $0.808 \pm 0.041$ | $0.692 \pm 0.033$ | $\mathbf{1.013 \pm 0.006}$ | $0.992 \pm 0.012$ |
| | Act./Global | $0.926 \pm 0.032$ | $0.679 \pm 0.023$ | $\mathbf{1.327 \pm 0.005}$ | $\mathbf{0.950 \pm 0.009}$ |
| | Act./Local | $0.956 \pm 0.030$ | $0.695 \pm 0.021$ | $\mathbf{1.379 \pm 0.004}$ | $\mathbf{0.930 \pm 0.008}$ |
| VGG-16, ImageNet | Weight/Global | $\mathbf{1.245 \pm 0.042}$ | $0.763 \pm 0.040$ | $\mathbf{1.043 \pm 0.005}$ | $0.998 \pm 0.001$ |
| | Weight/Local | $\mathbf{1.166 \pm 0.019}$ | $0.823 \pm 0.020$ | $\mathbf{1.076 \pm 0.003}$ | $\mathbf{0.991 \pm 0.001}$ |
| | Act./Global | $\mathbf{1.154 \pm 0.026}$ | $0.838 \pm 0.028$ | $\mathbf{1.066 \pm 0.003}$ | $1.000 \pm 0.001$ |
| | Act./Local | $\mathbf{1.084 \pm 0.020}$ | $0.879 \pm 0.021$ | $\mathbf{1.056 \pm 0.003}$ | $1.001 \pm 0.001$ |
| VGG-19, ImageNet | Weight/Global | | | $\mathbf{1.061 \pm 0.004}$ | $1.003 \pm 0.001$ |
| | Weight/Local | | | $\mathbf{1.099 \pm 0.003}$ | $1.001 \pm 0.001$ |
| | Act./Global | | | $\mathbf{1.046 \pm 0.003}$ | $\mathbf{0.996 \pm 0.001}$ |
| | Act./Local | | | $\mathbf{1.081 \pm 0.002}$ | $1.004 \pm 0.001$ |
| ResNet18, ImageNet | Weight/Global | $0.915 \pm 0.045$ | $0.957 \pm 0.047$ | | |
| | Weight/Local | $0.970 \pm 0.015$ | $0.970 \pm 0.016$ | | |
| | Act./Global | $0.972 \pm 0.016$ | $0.970 \pm 0.018$ | | |
| | Act./Local | $0.983 \pm 0.014$ | $0.967 \pm 0.015$ | | |

be with respect to input features, because if a single image can strongly excite an entire subcluster, this suggests that the neurons comprising it are involved in detecting/processing similar features. Second, we obtain a measure of coherence by analyzing the entropy $H(\text{label} \mid x(\theta, c); \theta)$ of the softmax outputs of the network when these images are passed through. If the entropy is low, this suggests that a cluster is coherent with respect to output labels.

Just as with lesion experiments, we perform analysis using these two methods using the pipeline from Section 3.2 to measure how coherent true subclusters are compared to random ones. Low Fisher-Bates $p$ values suggest the detection of an unexpectedly high number of coherent subclusters. For visualization score experiments, effect measures $> 1$ indicate coherence while for the softmax $H$ ones, effect measures $< 1$ indicate coherence. Figure 3 shows Fisher-Bates $p$ values, and Table 1 shows effect measure data. Figure 5 shows histograms of feature visualization percentiles for partitionings of a VGG network trained on CIFAR-10, to illustrate the significance of the Fisher-Bates $p$ values.

## 4.3 FINDINGS

**Our partitionings identify important subclusters.** Fisher-Bates $p$ values for lesion accuracy drops are low and significant, as shown in Figure 3, indicating that sub-clusters are more likely to be highly important relative to random groups of neurons. However, as shown in Table 1, not all of the corresponding effect measures are below one, even when the Fisher-Bates $p$ value is significant. This indicates that when we detect that an unusual number of subclusters are important, this does not necessarily correspond to importance on average.

**Our partitionings identify subclusters that are coherent w.r.t. input features but not class label.** Class-specific measures of coherence, class-wise lesion accuracy drop range and output entropy, showed significant coherence in almost no conditions. The class-wise range measure even tended to show that subclusters were less coherent w.r.t. class than random groups of neurons. However, the *visualization score* measure of coherence was reliably significant as shown in Figure 3, and the effect measures were reliably above 1. Together, these results offer evidence that subclusters tended to perform coherent sub-tasks, but not in a class-specific way.

**All partitioning methods give similar results.** In Figure 3, we find that activation-based clusterings have a weak tendency to produce lower Fisher-Bates $p$ values in our analysis compared to weight-based ones. Meanwhile, between local and global clustering methods, local methods seem to have a weak tendency to produce lower Fisher-Bates $p$ values. However, neither tendency was consistent, and all clustering methods performed similarly overall. This is somewhat unexpected: one might have predicted that weight-based methods' lack of runtime information or local methods' lack of global information would lead to lower quality clusterings, but this was not the case.

## 5 DISCUSSION

**Contributions:** In this work, we introduce several methods for partitioning networks into clusters of neurons and analyze the resulting partitions for modularity. Key to this is analyzing proxies: *importance* as a means of understanding what parts of a network are crucial for performance, and *coherence* as a measure for how specialized these parts are. We rigorously evaluate these proxies using statistical methods, finding that even the weights-only clustering methods are able to reveal clusters with a significant degree of importance and coherence compared to random ones. In each network, we found evidence that our partitioning methods were able to identify modular subsets of neurons via measuring accuracy drops under lesions (importance) and feature visualization scores (coherence). To the best of our knowledge, ours is the first method which is able to quantitatively assess modularity exhibited by neural networks in a way that does not require a human in the loop.

**Relation to other research:** Having effective tools for interpreting networks has been a key goal, especially when this aids in the diagnosis of failure modes (e.g., Carter et al. (2019); Mu & Andreas (2020)). Our work relates to this goal, though indirectly. The tests we perform are based on data from lesions and feature visualizations, both of which are interpretability tools. But rather than directly using these data to interpret subclusters, our focus is one step higher: on automatedly testing whether these subclusters are worth analyzing at all, and finding ways to screen for ones that should be the subject of deeper investigation. By showing that the partitioning methods we use identify subclusters that exhibit modularity, these results suggest that clustering neurons offers a useful level of abstraction through which to study networks.

**Limitations:** One limitation of our work is a lack of assurance that importance and coherence are reliably strong proxies for human-comprehensible forms of modularity. While they are sufficient to imply some degree of abstractability with respect to the task at hand, they may not always be particularly useful for a prosaic understanding. Relatedly, our approach is also not designed to identify the subtask performed by a set of units which exhibit modularity, nor does it identify relationships between modules. Given these limitations, the tools we introduce should be seen as methods for screening a network in search for evidence of modularity overall and for particular sets of units which seem to exhibit it and merit additional investigation. A final notable limitation is that these methods do not offer tools for building more modular networks beyond techniques for measuring it. Future work toward this may benefit from our techniques but should also hinge on architectural or regularization-based methods for promoting modularity in networks.

**Conclusion:** While we make progress here toward better understanding how networks can be understood, neural systems are still complex, and more insights are needed to develop prosaic understandings of them. The ultimate goal would be to develop reliable methods for building models which are both dynamic and lend themselves to faithful semantic interpretation. We believe that using modularity as an organizing principle and expanding our interpretability toolbox with more modular models and better-tuned partitioning methods will lead to a richer understanding of networks and better tools for building them to be reliable and verifiable.

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

## A   APPENDIX

### A.1   SPECTRAL CLUSTERING ALGORITHM

The spectral clustering algorithm on the graph $G = (V, E)$ produces a partition of its vertices in which there are stronger connections within sets of vertices than between them (Shi & Malik, 2000). It does so by solving a relaxation of the NP-Hard problem of minimizing the *n-cut* (normalized cut) for a partition. For disjoint, non-empty sets $X_1, ...X_k$ where $\cup_{i=1}^{k} X_i = V$, this is defined by von Luxburg (2007) as:

$$\text{n-cut}(X_1, ..., X_k) := \frac{1}{2} \sum_{i=1}^{k} \frac{W(X_i, \overline{X_i})}{\text{vol}(X_i)}$$

for two sets of vertices $X, Y \subseteq V$, we define $W(X, Y) := \sum_{v_i \in X, v_j \in Y} w_{ij}$; the degree of a vertex $v_i \in V$ is $d_i = \sum_{j=1}^{n} w_{ij}$; and the volume of a subset $X \subseteq V$ is $\text{vol}(X) := \sum_{i \in X} d_i$.

We use the *scikit-learn* implementation (Pedregosa et al., 2011) with the ARPACK eigenvalue solver (Borzì & Borzì, 2006).

---

**Algorithm 1: Normalized spectral clustering** according to Shi & Malik (2000), implemented in *scikit-learn* (Pedregosa et al., 2011), description taken from von Luxburg (2007).

**Input**   : Weighted adjacency matrix $W \in \mathbb{R}^{n \times n}$, number $k$ of clusters to construct
1  Compute the unnormalized Laplacian $L$.
2  Compute the first $k$ generalized eigenvectors $u_1, ..., u_k$ of the generalized eigenproblem $Lu = \lambda Du$.
3  Let $U \in \mathbb{R}^{n \times k}$ be the matrix containing the vectors $u_1, ..., u_k$ as columns.
4  For $i = 1, .., n$, let $y_i \in \mathbb{R}^k$ be the vector corresponding to the $i^{\text{th}}$ row of $U$.
5  Cluster the points $(y_i)_{i=1,...,n}$ in $\mathbb{R}^k$ with the *k-means* algorithm into clusters $C_1, ..., C_k$,
**Output** : Clusters $A_1, ..., A_k$ with $A_i = \{j | y_j \in C_i\}$.

---

### A.2   NETWORK TRAINING DETAILS

We use Tensorflow's implementation of the Keras API (Abadi et al., 2015; Chollet et al., 2015). When training all networks, we use the Adam algorithm (Kingma & Ba, 2014) with the standard Keras hyperparameters: learning rate 0.001, $\beta_1 = 0.9$, $\beta_2 = 0.999$, no amsgrad. The loss function was categorical cross-entropy. For all non-ImageNet networks, we train 5 identically-configured replicates.

**Small MLPs (MNIST):** We train MLPs with 4 hidden layers, each of width 256, for 20 epochs with batch size 128. All MLPs achieved a test accuracy on MNIST between 97.6% and 98.2%. On Fashion-MNIST, they achieved test accuracies between 88.4% and 89.4%.

**Small CNNs (MNIST):** These networks had 3 convolutional layers with 64 $3 \times 3$ channels each with the second and third hidden layers being followed by max pooling with a 2 by 2 window. There was a final fully-connected hidden layer with 128 neurons. We train them with a batch size of 64 for 10 epochs. All small CNNs achieved a testing accuracy MNIST between 99.1% and 99.4%. On Fashion-MNIST, they achieved test accuracies between 91.8% and 92.4%.

**Mid-sized VGG CNNs (CIFAR-10):** We implement a version of VGG-16 described by Simonyan & Zisserman (2014); Liu & Deng (2015). We train these with Adam, for 200 epochs with a batch size of 128. These are trained using $L_2$ regularization with a coefficient of $5 \times 10^{-4}$ and dropout with a rate tuned per-layer as done in Simonyan & Zisserman (2014); Liu & Deng (2015). Training was done with data augmentation which consisted of random rotations between 0 and 15 degrees, random shifts both vertically and horizontally of up to 10% of the side length, and random horizontal flipping. The unregularized networks achieved testing accuracies between 90.3% and 90.9% while the regularized ones did between 82.8% and 86.6%.

**Large CNNs (ImageNet):** We experimented with VGG-16 and 19 (Simonyan & Zisserman, 2014) and ResNet-18, and 50 (He et al., 2016) networks. Weights were obtained from the Python `image-classifiers` package, version 1.0.0.

## A.3 PIPELINE - SECOND PART

Figure 4: **Our extended procedural pipeline.** This figure expands Figure 2 and shows the successive steps after generating a partitioning of subclusters (step 3 in Figure 2). After performing either lesion or feature visualization analysis, the results from each true cluster and its random clusters are aggregated to produce $p$ values and effect measures. For simplicity, only the analysis for the lesion experiment is presented, but the same pipeline is used for the feature vis experiments.

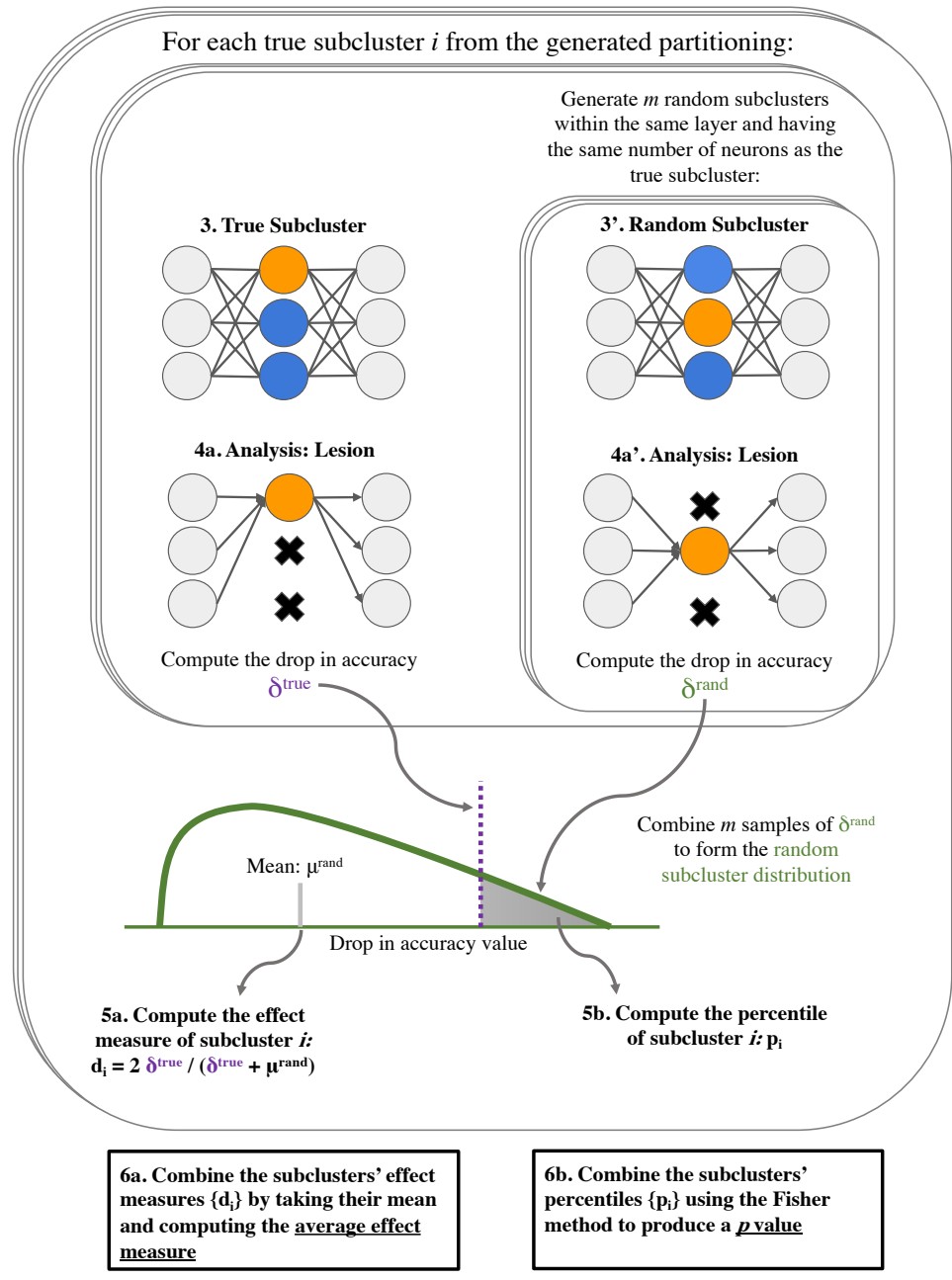

Figure 5: **Histograms of feature visualization percentiles for four partitionings of VGG CIFAR-10 network.** A VGG CIFAR-10 network is partitioned using four methods ({weights, activations} × {global, local}) and analyzed with the feature visualization experiment to produce the collection of percentiles for each subcluster. This figure shows the percentile distribution for each clustering as a histogram with a single percentile value per bin. Recall that a lower percentile means that a true subcluster is disproportionately often more coherent than random subclusters while controlling for layer and size. Under the null hypothesis, percentiles distribute uniformly. Visual inspection demonstrates that, for example, Weights / Global, Act / Global, and Act / Local deviate from the uniform distribution. In this paper, we use the Fisher-Bates procedure to conduct this hypothesis testing rigorously. Indeed, Figure 3 shows that these clustering of the VGG CIFAR-10 network are significant when aggregated over five models.

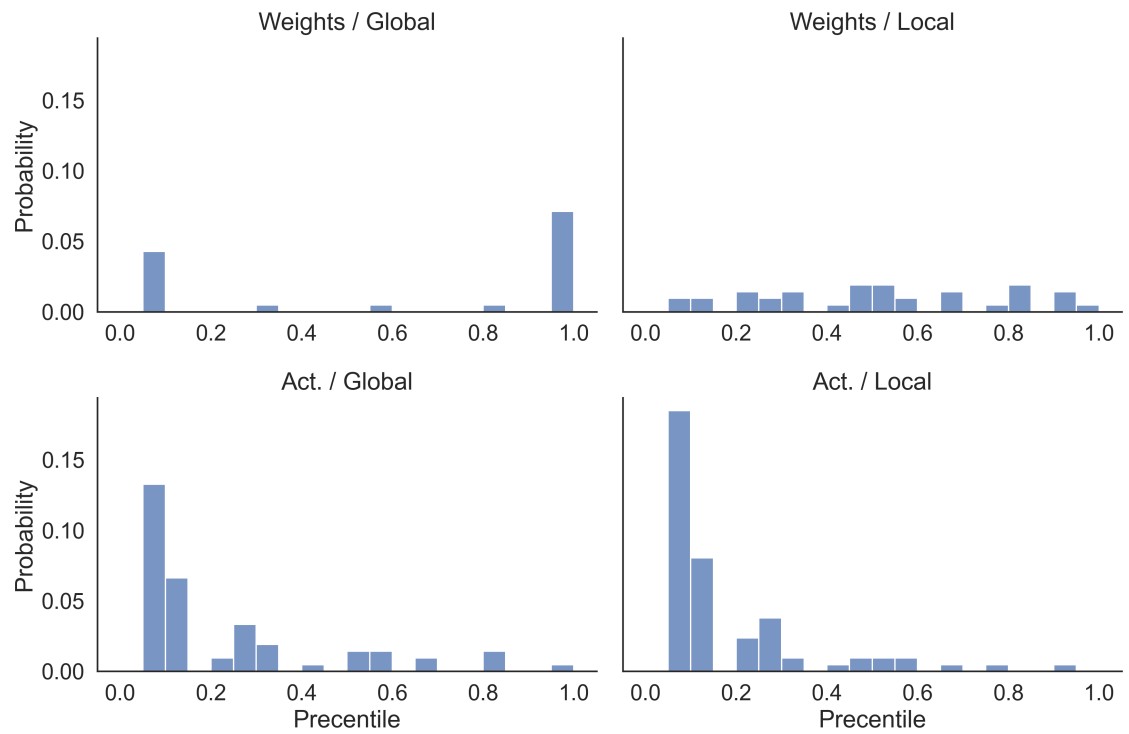

## A.4 Correlation-Based Visualization

**Halves-MNIST Dataset:** Figure 6 shows examples from our halves-MNIST dataset. To create each example, two images were randomly selected, resized to have half their original width, and concatenated together. Each image was labeled with the sum of the two digits modulo 10. Examples are shown in Figure 1.

**Visualization:** To create the images from Figure 1 which show clusters of neurons in the first layer of an MLP trained on halves-MNIST, we use a correlation-based visualization algorithm from Watanabe (2019). We construct visualizations of neurons using their correlations with the input pixels' pre ReLU activities across the test dataset. Instead of Pearson (linear) correlation, we use the Spearman correlation (which is the Pearson correlation of ranks) because it is able to capture relationships which tend to monotonically increase even if they are nonlinear.

After obtaining visualizations for each neuron in a subcluster, we do not directly take their average to visualize the entire subcluster. To see why, consider two neurons which are highly anticorrelated across the testing set. These neurons are highly coherent, but averaging together their visualizations would obscure this by cancellation. To fix this problem, we align the signs of the visualizations for individual neurons using a variant of a stoahcstic alignment algorithm from Watanabe (2019). To visualize a subcluster, for a number of iterations (we use 20), we iterate over its neurons and calculate

for each the sum of cosines between its visualization and each of the other neurons' visualizations in vector form. If this sum is negative, we flip the sign of this neuron's visualization. After this procedure, we take the mean of the visualizations within a subcluster. This process is detailed in Algorithm 2.

---

**Algorithm 2:** Sign Alignment Algorithm (Similar to Watanabe (2019))

---

**Result:** Set of sign-aligned neuron visualizations.
**Input** Neuron visualizations $V_{1:n}$ **for** *iter in num_iters* **do**
    **for** $v_i$ *in* $V$ **do**
        Calculate sum of cosines, $c = \sum_{j \neq i} \frac{v_i \cdot v_j}{\sqrt{v_i \cdot v_i} \sqrt{v_j \cdot v_j}}$
        **if** $c < 0$ **then**
            $v_i \leftarrow -v_i$
        **end**
    **end**
**end**

---

**Comparison to Random Subclusters:** To confirm that the visualization in Figure 1 show that our clustering algorithms are capturing modularity in the first layer of the MLPs, we compare them to the same visualizations but for random subclusters in Figure 6. The visualizations that were used to produce Figure 1 are each at the top of a column of one of the panels in Figure 6 and are above visualizations for random subclusters of the same size and layer. Each column in each panel was independently scaled to have values in the interval [0,1]. In each of the four panels, visualizations in the first row reflect the most selectivity to one side or the other meaning that compared to random subclusters, the ones found via these clustering methods tend to be significantly more coherent.

**MLPs can Compute Modular Sums:** A network an do this for $M$ values by using an intermediate layer of $M^2$ neurons, each of which serve as a detector of one of the possible combinations of inputs. Consider a ReLU MLP with $2M$ inputs, a single hidden layer with $M^2$ neurons, and then $M$ outputs. Suppose that it is given the task of mapping datapoints in which the input nodes numbered $i$ and $M + j$ are activated with value 1 to an output in which the $\mod (i + j, M)^{\text{th}}$ node is active with value 1. It could do so if each hidden neuron with a ReLU activation detected one of the $M^2$ possible input combinations via a bias of -1 and two weights of 1 connecting it to each of the input nodes in the combination is detects. A single weight from each hidden neuron to its corresponding output point would allow the network to compute the modular sum. In our MLPs, we have $M = 10$ classes, and the MLPs have a dense layer with $> 10^2$ neurons preceding the output layer. Thus, they are capable of computing a modular sum in the halves and stack-diff tasks we give to them.

## A.5 FEATURE VISUALIZATION

All visualizations were created using the Lucid[10] package. The optimization objective for visualizing sub-clusters was the $L_1$ norm of the pre-ReLU inputs for all neurons inside the subcluster (it was an $L_1$ norm of $L_1$ norms for convolutional feature maps). For small MLPs, small CNNs, and mid-sized CNNs, we generated images using random jittering and scaling, and for ImageNet models, we used Lucid's default transformations which consist of padding, jittering, rotation, and scaling with default hyperparameters. For all networks, we used the standard pixel-based parameterization of the image and no regularization on the Adam optimizer for 100 steps. For visualizations in small MLPs and CNNs, we used versions of these networks trained on 3-channel versions of their datasets in which the same inputs were stacked thrice because Lucid requires networks to have 3-channel inputs. However, we show grayscaled versions of these in figure 7. Refer to Section 4 of the main paper for quantitative analysis of the optimization objective values.

## A.6 ROBUSTNESS TO THE CHOICE OF CLUSTER NUMBER

In the main paper, we only present results form experiments in which the number of clusters, $k$, was set to 16. The fact that we find evidence of modularity in networks from the MNIST to ImageNet scale using $k = 16$ suggests that detecting it is robust to $k$. However, here we also present a direct

---

[10] https://github.com/tensorflow/lucid

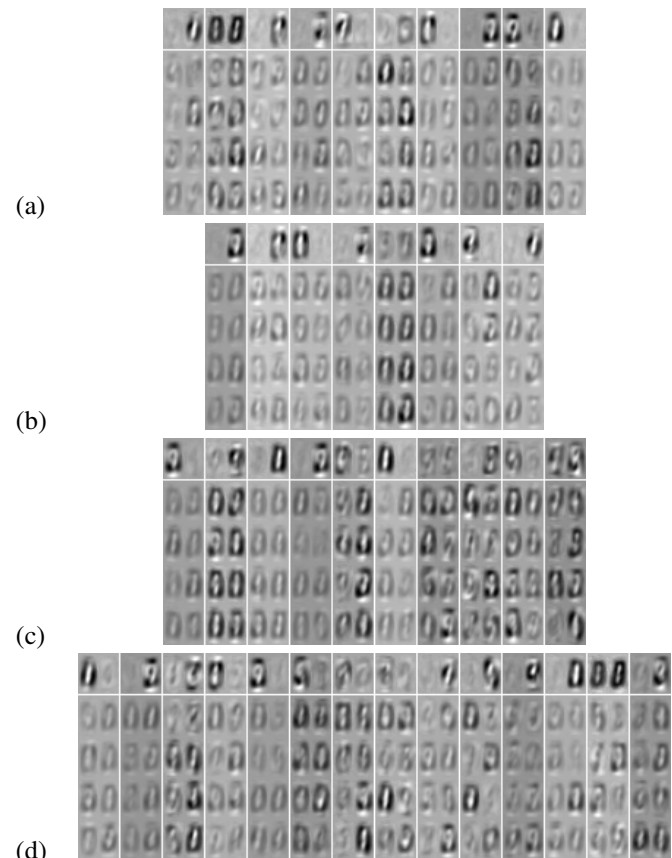

(a)

(b)

(c)

(d)

Figure 6: Comparison of true and random subcluster visualizations for the first layer of MLPs trained on the halves-MNIST dataset. The first rows show visualizations for true subclusters, and the bottom four show visualizations for random ones of the same size. Each panel gives results for a different approach to clustering: (a) weights/global, (b) weights/local, (c) activity/global, and (d) activity/local.

comparison between results for CIFAR CNN-VGGs for $k \in \{8, 12, 16\}$. Tables 2 and 3 show these results for lesion and feature-visualization experiments respectively. In general, whether or not an experiment resulted in a significant Fisher-Bates $p$ value or an effect measure suggesting modularity is consistent across these values of $k$.

## A.7 MULTIPLE TESTING CORRECTIONS

In the main paper in Tables 4 and 5, we report various Fisher-Bates $p$ values that summarize the degree to which statistics of sub-clusters vary from those of random groups of neurons within a network. For each network, the Fisher-Bates $p$ values one-sidedly test whether the true sub-cluster measures reflect more importance/coherence than those of random subclusters. However, when testing multiple networks, one may want to reduce the chance of false positives due to the sheer number of tests performed. To incorporate this analysis into our results, we perform a multiple testing correction within Tables 4 and 5 (separately) using the Benjamini-Hochberg procedure (Benjamini & Hochberg, 1995). This procedure controls the false discovery rate: that is, the expected proportion of rejections of the null hypothesis that are false positives, where the expectation is taken under the data-generating distribution. It relies on all experiments being independent. For a false discovery rate $\alpha$ and $m$ ordered $p$ values $\{p_1...p_m\}$, this procedure chooses a critical $p$ value as $p_k$ where $k = \arg\max_{j \in 1:m} \mathbb{I}(p_j \leq \frac{j\alpha}{m})$ where $\mathbb{I}$ is an indicator. All $p$ values greater than $p_k$ are deemed not significant at this level. In Figure 3 we demarcate $p$ values significant at the $\alpha = 0.05$ level under this correction. In our case, the critical value was $p_k = 0.025$.

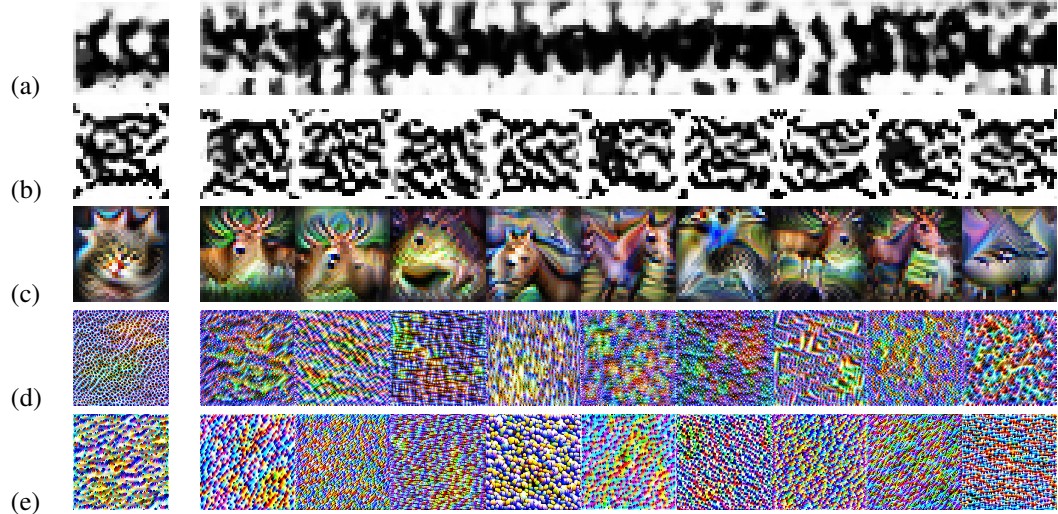

Figure 7: **Example feature visualizations for true and random sub-clusters:** In the left column are shown true sub-cluster visualizations, and in the right column are visualizations of sub-clusters of random neurons of the same size in the same layer. (a) MLP, MNIST; (b) CNN, MNIST; (c) CNN-VGG, CIFAR-10; (d) VGG-16, ImageNet; (e) VGG-19, ImageNet.

Table 2: Comparison for different cluster number values in CNN-VGG networks for lesion-based experiments. Results are shown for $k \in \{8, 12, 16\}$. Each Fisher-Bates $p$ value which is significant under the Benjamini Hochberg multiple correction at $\alpha = 0.05$ is **bold**. An effect measure $> 1$ corresponds to more importance/coherence among true subclusters than random ones. All above 1 are **bold**.

| Network k | Partitioning | Importance: Acc. Drop | | Coherence: Class Range | |
|---|---|---|---|---|---|
| | | p Value | Effect Meas. High→Imp. | p Value | Effect Meas. High→Coh. |
| VGG, CIFAR-10 $k = 8$ | Weight/Global | $\mathbf{2.33 \times 10^{-12}}$ | 0.912 | $\mathbf{4.25 \times 10^{-14}}$ | 0.430 |
| | Weight/Local | 0.266 | 0.851 | **0.021** | 0.869 |
| | Act./Global | $\mathbf{2.55 \times 10^{-10}}$ | 0.969 | 0.578 | 0.764 |
| | Act./Local | $\mathbf{1.06 \times 10^{-21}}$ | **1.055** | 0.866 | 0.772 |
| VGG, CIFAR-10 $k = 12$ | Weight/Global | $\mathbf{4.06 \times 10^{-8}}$ | 0.739 | $\mathbf{1.64 \times 10^{-15}}$ | 0.632 |
| | Weight/Local | 0.204 | 0.873 | 0.101 | 0.795 |
| | Act./Global | $\mathbf{2.70 \times 10^{-13}}$ | 0.903 | 0.134 | 0.741 |
| | Act./Local | $\mathbf{6.30 \times 10^{-17}}$ | 0.994 | 0.759 | 0.720 |
| VGG, CIFAR-10 $k = 16$ | Weight/Global | $\mathbf{4.40 \times 10^{-8}}$ | 0.682 | $\mathbf{2.44 \times 10^{-29}}$ | 0.407 |
| | Weight/Local | **0.021** | 0.808 | $\mathbf{2.24 \times 10^{-4}}$ | 0.692 |
| | Act./Global | $\mathbf{1.62 \times 10^{-11}}$ | 0.926 | 0.845 | 0.679 |
| | Act./Local | $\mathbf{2.26 \times 10^{-23}}$ | 0.956 | 0.266 | 0.695 |

## A.8 FULL TABULAR DATA

Tables 4 (lesion experiments data) and 5 (feature visualization experiment data) show the $p$ value results from Figure 3 and effect measure results from Table 1 but do so in a form that places the Fisher-Bates $p$ value and effect measure data from each network in the same row. All Fisher-Bates $p$ values significant according to the Benjamini Hochberg method are in **bold**, and all effect measures indicating greater average importance/coherence among true subclusters compared to random subclusters are in **bold**.

Table 3: Comparison for different cluster values in CNN-VGG networks for feature visualization-based experiments. Tesults are shown for $k \in 8, 12, 16$. Each Fisher-Bates $p$ value which is significant at $\alpha = 0.05$ level under the Benjamini Hochberg multiple correction is **bold**. For vis score tests, an effect measure $> 1$ corresponds to more coherence, and for softmax $H$ tests, one of $< 1$ corresponds to more coherence. All effect measures on the size of 1 indicating more coherence are **bold**

| Network k | Partitioning | Coherence: Vis Score p Value | Effect Meas. High→Coh. | Coherence: Softmax H p Value | Effect Meas. Low→Coh. |
|---|---|---|---|---|---|
| VGG, CIFAR-10 $k = 8$ | Weight/Global | $\mathbf{5.85 \times 10^{-12}}$ | **1.011** | 0.032 | 1.028 |
| | Weight/Local | 0.939 | 0.974 | 0.784 | 1.059 |
| | Act./Global | $\mathbf{1.58 \times 10^{-57}}$ | **1.345** | 0.368 | **0.940** |
| | Act./Local | $\mathbf{8.11 \times 10^{-70}}$ | **1.358** | 0.179 | **0.946** |
| VGG, CIFAR-10 $k = 12$ | Weight/Global | $\mathbf{3.71 \times 10^{-9}}$ | 0.959 | 0.917 | 1.106 |
| | Weight/Local | 0.732 | 0.993 | 0.089 | **0.945** |
| | Act./Global | $\mathbf{2.96 \times 10^{-78}}$ | **1.363** | 0.065 | **0.917** |
| | Act./Local | $\mathbf{1.57 \times 10^{-102}}$ | **1.364** | 0.041 | **0.928** |
| VGG, CIFAR-10 $k = 16$ | Weight/Global | $\mathbf{6.98 \times 10^{-8}}$ | 0.871 | 0.613 | 1.124 |
| | Weight/Local | 0.442 | **1.013** | 0.616 | **0.992** |
| | Act./Global | $\mathbf{5.59 \times 10^{-64}}$ | **1.327** | 0.045 | **0.950** |
| | Act./Local | $\mathbf{7.79 \times 10^{-121}}$ | **1.379** | **0.013** | **0.930** |

Table 4: Results for lesion-based experiments in networks involving importance as measured through overall accuracy drops and coherence as measured by the class-wise range of accuracy drops. Each row corresponds to a network paired with a partitioning method. Results are calculated as explained in Section 3.2. Each Fisher-Bates $p$ value which is significant at an $\alpha = 0.05$ level under the Benjamini Hochberg multiple correction is in **bold**. In this case, significance means $p \leq 0.025$. For both accuracy drop and classwise range experiments, an effect measure $> 1$ corresponds to more importance/coherence among true subclusters than random ones. All such effect measures which are further than two standard errors above 1 are in **bold**.

| Network | Partitioning | Importance: Acc. Drop p Value | Effect Meas. High→Imp. | Coherence: Class Range p Value | Effect Meas. High→Coh. |
|---|---|---|---|---|---|
| MLP, MNIST | Weight/Global | $\mathbf{1.19 \times 10^{-17}}$ | $\mathbf{1.123 \pm 0.058}$ | 0.817 | $0.701 \pm 0.036$ |
| | Weight/Local | $\mathbf{5.18 \times 10^{-27}}$ | $1.061 \pm 0.048$ | 0.914 | $0.676 \pm 0.029$ |
| | Act./Global | $\mathbf{2.36 \times 10^{-11}}$ | $0.883 \pm 0.038$ | 0.556 | $0.646 \pm 0.024$ |
| | Act./Local | $\mathbf{5.80 \times 10^{-22}}$ | $0.929 \pm 0.040$ | 0.315 | $0.687 \pm 0.025$ |
| CNN, MNIST | Weight/Global | $\mathbf{1.64 \times 10^{-5}}$ | $0.837 \pm 0.048$ | 0.984 | $0.527 \pm 0.031$ |
| | Weight/Local | 0.063 | $0.814 \pm 0.046$ | 0.821 | $0.635 \pm 0.033$ |
| | Act./Global | $\mathbf{3.52 \times 10^{-10}}$ | $1.078 \pm 0.061$ | 1.000 | $0.543 \pm 0.039$ |
| | Act./Local | $\mathbf{1.56 \times 10^{-7}}$ | $0.939 \pm 0.060$ | 0.841 | $0.625 \pm 0.043$ |
| VGG, CIFAR-10 | Weight/Global | $\mathbf{4.40 \times 10^{-8}}$ | $0.682 \pm 0.066$ | $\mathbf{2.44 \times 10^{-29}}$ | $0.407 \pm 0.042$ |
| | Weight/Local | **0.021** | $0.808 \pm 0.041$ | $\mathbf{2.24 \times 10^{-4}}$ | $0.692 \pm 0.033$ |
| | Act./Global | $\mathbf{1.62 \times 10^{-11}}$ | $0.926 \pm 0.032$ | 0.845 | $0.679 \pm 0.023$ |
| | Act./Local | $\mathbf{2.26 \times 10^{-23}}$ | $0.956 \pm 0.030$ | 0.266 | $0.695 \pm 0.021$ |
| VGG16, ImageNet | Weight/Global | $\mathbf{9.44 \times 10^{-26}}$ | $\mathbf{1.245 \pm 0.042}$ | 1.000 | $0.763 \pm 0.040$ |
| | Weight/Local | $\mathbf{2.35 \times 10^{-52}}$ | $\mathbf{1.166 \pm 0.019}$ | 1.000 | $0.823 \pm 0.020$ |
| | Act./Global | $\mathbf{1.84 \times 10^{-25}}$ | $\mathbf{1.154 \pm 0.026}$ | 1.000 | $0.838 \pm 0.028$ |
| | Act./Local | $\mathbf{1.55 \times 10^{-27}}$ | $\mathbf{1.084 \pm 0.020}$ | 1.000 | $0.879 \pm 0.021$ |
| ResNet18, ImageNet | Weight/Global | **0.002** | $0.915 \pm 0.045$ | 0.061 | $0.957 \pm 0.047$ |
| | Weight/Local | $\mathbf{3.57 \times 10^{-6}}$ | $0.970 \pm 0.015$ | 0.743 | $0.970 \pm 0.016$ |
| | Act./Global | $\mathbf{5.94 \times 10^{-6}}$ | $0.972 \pm 0.016$ | 0.956 | $0.970 \pm 0.018$ |
| | Act./Local | $\mathbf{1.18 \times 10^{-10}}$ | $0.983 \pm 0.014$ | 0.958 | $0.967 \pm 0.015$ |

Table 5: Results for feature visualization-based experiments in networks involving coherence as measured by the optimization score of feature visualizations and the entropy of network outputs. Each row corresponds to a network paired with a partitioning method. Results are calculated as explained in Section 3.2. Each Fisher-Bates $p$ value which is significant at an $\alpha = 0.05$ level under the Benjamini Hochberg multiple correction is in **bold**. In this case, significance means $p \leq 0.025$. For visualization score experiments, an effect measure $> 1$ corresponds to more coherence among true subclusters than random ones, while one of $< 1$ does for softmax entropy experiments. All effect measures which are more than two standard errors away from 1 on the size reflecting greater coherence among true subclusters are in **bold**

| Network | Partitioning | Coherence: Vis Score | | Coherence: Softmax H | |
|---|---|---|---|---|---|
| | | p Value | Effect Meas. High→Coh. | p Value | Effect Meas. Low→Coh. |
| MLP, MNIST | Weight/Global | **0.001** | $1.003 \pm 0.004$ | 0.707 | $1.105 \pm 0.021$ |
| | Weight/Local | **0.002** | $\mathbf{1.024 \pm 0.004}$ | **0.023** | $0.931 \pm 0.016$ |
| | Act./Global | $\mathbf{4.45 \times 10^{-5}}$ | $\mathbf{1.020 \pm 0.003}$ | **0.025** | $0.997 \pm 0.013$ |
| | Act./Local | $\mathbf{8.94 \times 10^{-7}}$ | $\mathbf{1.026 \pm 0.003}$ | **0.013** | $1.011 \pm 0.012$ |
| CNN, MNIST | Weight/Global | 0.082 | $0.998 \pm 0.004$ | 0.863 | $1.026 \pm 0.008$ |
| | Weight/Local | 0.344 | $1.004 \pm 0.003$ | 0.519 | $1.007 \pm 0.007$ |
| | Act./Global | 0.266 | $0.933 \pm 0.006$ | 0.831 | $1.025 \pm 0.011$ |
| | Act./Local | 0.219 | $0.925 \pm 0.005$ | 0.668 | $\mathbf{0.970 \pm 0.010}$ |
| VGG, CIFAR-10 | Weight/Global | $\mathbf{6.98 \times 10^{-8}}$ | $0.871 \pm 0.011$ | 0.613 | $1.124 \pm 0.012$ |
| | Weight/Local | 0.442 | $\mathbf{1.013 \pm 0.006}$ | 0.616 | $0.992 \pm 0.012$ |
| | Act./Global | $\mathbf{5.59 \times 10^{-64}}$ | $\mathbf{1.327 \pm 0.005}$ | 0.045 | $\mathbf{0.950 \pm 0.009}$ |
| | Act./Local | $\mathbf{7.79 \times 10^{-121}}$ | $\mathbf{1.379 \pm 0.004}$ | **0.013** | $\mathbf{0.930 \pm 0.008}$ |
| VGG-16, ImageNet | Weight/Global | $\mathbf{5.50 \times 10^{-8}}$ | $\mathbf{1.043 \pm 0.005}$ | 0.052 | $0.998 \pm 0.001$ |
| | Weight/Local | $\mathbf{2.52 \times 10^{-39}}$ | $\mathbf{1.076 \pm 0.003}$ | 0.155 | $\mathbf{0.991 \pm 0.001}$ |
| | Act./Global | $\mathbf{2.65 \times 10^{-15}}$ | $\mathbf{1.066 \pm 0.003}$ | 0.208 | $1.000 \pm 0.001$ |
| | Act./Local | $\mathbf{1.36 \times 10^{-34}}$ | $\mathbf{1.056 \pm 0.003}$ | 0.633 | $1.001 \pm 0.001$ |
| VGG-19, ImageNet | Weight/Global | $\mathbf{1.87 \times 10^{-10}}$ | $\mathbf{1.061 \pm 0.004}$ | 0.398 | $1.003 \pm 0.001$ |
| | Weight/Local | $\mathbf{3.69 \times 10^{-81}}$ | $\mathbf{1.099 \pm 0.003}$ | 0.458 | $1.001 \pm 0.001$ |
| | Act./Global | $\mathbf{6.01 \times 10^{-22}}$ | $\mathbf{1.046 \pm 0.003}$ | 0.089 | $\mathbf{0.996 \pm 0.001}$ |
| | Act./Local | $\mathbf{2.63 \times 10^{-27}}$ | $\mathbf{1.081 \pm 0.002}$ | 0.645 | $1.004 \pm 0.001$ |

