# OpenReview forum: "Detecting Modularity in Deep Neural Networks"
_ICLR.cc/2022/Conference — ICLR 2022 Submitted_

### Official Review · Reviewer_hnkg · 2021-10-31

**Correctness:** 4
**Technical Novelty And Significance:** 3
**Empirical Novelty And Significance:** 3
**Recommendation:** 6
**Confidence:** 2

**Main Review:**

Strengths:
1. This paper is very well written and easy to follow. The methods and results are convincing and clear.
2. This paper carefully designed gratification methods for neural network neurons and proposed plausible methods for not only global clusters and also local clusters.
3. They also provide a statistical method to identify the significant modules automatically. Though the power is very limited.

Some comments (not exactly weakness):
1. The global gratification is very interesting. I am wondering if the author looked into the special cases of clustering. For example, is there any chance that the global cluster is from disconnected layers? Specifically, a cluster that has neurons from layer 1 and layer 3 but does not have layer 2  neurons.
2. What is the purpose of local clustering? Why the clustering number of local clustering has to be the same as the global clustering. And how to explain the discrepancy between local clustering and global clustering.
3. The overall gratification method is somewhat heuristic. And I assume that is the reason why the p-value does not work. Because the gratification did not capture the very "true" connections of each neuron. But identifying such a "true" connection will be very difficult. I was wondering could the author consider using the unit interference of each neuron and test the impact of other neurons. And using the impact to reflect the connections of the neurons.
4. Does the identified modules can be taken advantage of to train the neuron network in some way. Links to lottery tickets?
5. Did the author consider other graph clustering methods? Like Louvain community detection, hierarchical clustering?





**Summary Of The Paper:**

This paper uses graph-based clustering methods to identify the humanly comprehensible modularities in the neural network. They thoroughly discussed how they construct the graph of neurons and carry out the clustering. They also proposed the fisher bates p-value for testing the significance of identified modules.

**Summary Of The Review:**

A good paper to explore the explainability of neuron networks with graph clustering methods.

---

> ### Author Response · Authors · 2021-11-16
> **Response to hnkg**
>
> Thank you for the constructive feedback.
>
> 1. Re: “The global gratification is very interesting. I am wondering if the author looked into the special cases of clustering. For example, is there any chance that the global cluster is from disconnected layers? Specifically, a cluster that has neurons from layer 1 and layer 3 but does not have layer 2 neurons.”
>
> This is possible for activation-based clusterings, though we find that the clusters span all or almost all layers in most cases. However, this would never happen for weight-based clusterings in nondegenerate cases because the neurons in such a cluster would not share any weights. This relates to one of the major differences between global weight and global activation-based clusterings. Weight-based clusterings tend to span several adjacent layers, while ones from activation-based clusterings tend to span most layers of the network.
>
> Note that for the modularity evaluation using the importance and coherence proxies, we use only subclusters (the intersection of cluster and layer; to control for layer representation.).
>
> 2. Re: “What is the purpose of local clustering? Why the clustering number of local clustering has to be the same as the global clustering. And how to explain the discrepancy between local clustering and global clustering.”
>
> Local clustering is simpler and faster to run than global clustering because it requires the construction of much smaller graphs. We find it interesting and convenient that it is not worse at identifying modularity than global clustering in our experiments. We control the number of clusters per layer to ensure comparability between the local and global clusterings in our experiments, but this would not be done for any practical application. In general, the discrepancy is small between local and global clustering results, with no consistent trend across results. This suggests that we may not need information about far away layers to reliably determine how much neurons in one layer relate to each other.
>
>
> 3. Re: “The overall gratification method is somewhat heuristic. And I assume that is the reason why the p-value does not work. Because the gratification did not capture the very ‘true’ connections of each neuron. But identifying such a ‘true’ connection will be very difficult. I was wondering could the author consider using the unit interference of each neuron and test the impact of other neurons. And using the impact to reflect the connections of the neurons.”
>
> This would certainly be a way to do graphification--probably a good one. But ultimately, we stick to our 4 spectral clustering methods because it is well-fit to the problem, thoroughly studied, scalable to ImageNet, and previously used by Filan et al. (2021). We could try others, but our goal has not been to compare many graphification methods but to demonstrate how modularity could be discovered (importance and coherence proxies). We don't think this would be of comparable interest to the field.
>
> 4. Re: “Did the author consider other graph clustering methods? Like Louvain community detection, hierarchical clustering?”
>
> We have not, but as in the previous comment, our goal has not been to compare other graph clustering methods.
>
> 5. Re: “Does the identified modules can be taken advantage of to train the neuron network in some way. Links to lottery tickets?”
>
> Filan et al. (2021) experiment with two ways to train networks to be more spectrally clusterable. Others could be devised as well such as [softly] supervising the internal activations of neurons in a network. We do not find any particularly strong connections between our work and lottery tickets.

---

### Official Review · Reviewer_ZAxP · 2021-11-02

**Correctness:** 3
**Technical Novelty And Significance:** 2
**Empirical Novelty And Significance:** 3
**Recommendation:** 5
**Confidence:** 4

**Main Review:**

The aim of this work is important and surely relevant. I find it likely that some degree of modularity exists in networks and that it is possible to empirically detect their existence.

However I have some doubts on the methodology that is proposed in this paper.

1) It is not immediately obvious that the 5 steps described above really provide a good indication of modularity.  As a counter example, imagine to train a network with most nodes masked to zero. In this scenario, at the end of training only a small group of nodes will be ‘active’ (connected by nonzero weights), and such group of nodes will be hugely more ‘important’ and ‘coherent’ than random sets of nodes, but this is *not* an indication of the network modularity. What is the authors’ opinion on this?

2) Connected to the first point, I see an issue in the high level of arbitrariness in the definition of modularity given. In particular, for the scheme one needs to choose: the graph construction algorithm, the clustering algorithm and its parameters and the definition of importance and coherence scores. All these choices will give rise to *different modules* and I see no way in which one should prefer one scheme with respect to the other.

3) I think that all the heuristic choices described in point 2 make the method difficult to reproduce and to extend, and hence there is a risk for this work to become an isolated publication generating little attention and very few further investigations.

3) What is the value in knowing that some 'modules' exist if we can never discover what these modules do in practice? The article would be hugely more impactful if the authors could provide an indication of the functioning of the modules detected. I would appreciate if the author could comment (both here and in the paper) on whether this could be done at all and, in case, on the possible ways in which it could be done.



**Summary Of The Paper:**

In this work the authors provide an empirical definition of 'modularity' of trained deep networks. Their definition is ‘automated’ in the sense that is does not require any human identification and evaluation of the specific modules.

The proposed definition consists in the following steps.

1) Construct an undirected graph from the nodes and weights of the trained-network

2) Divide the graph in groups using a spectral clustering algorithm

3)  Define ‘subclusters’ as the sets of all nodes that are in the same cluster and in the same layer

4) For each subcluster compute an *importance* score and a *coherence* score

5) If the sub-clusters are more important and coherent than randomly defined groups of nodes of the same size than the network is said to be ‘modular’.

The *importance* of a group of nodes is computed as the loss of accuracy that results from masking to zero that group of nodes. The greater the loss of accuracy, the greater the importance of the group.

The *coherence* of a group of nodes is computed as the maximum magnitude of the pre-activation weights of the group that is achievable with a single image.

By numerical testing, the authors find that it is indeed possible to find groups of nodes that are more important and coherent than random in a statistically significant way for different architectures.


**Summary Of The Review:**

Detecting modularity in networks is surely an important and timely problem in deep learning. The authors proposal to address it contains some interesting elements of novelty, but I am uncertain that the method really measures modularity and I believe that the arbitrariness and the many heuristics involved in the definition of modularity make the methodology difficult to trust, reproduce, or extend.

---

> ### Author Response · Authors · 2021-11-16
> **Response to ZAxP**
>
> Thank you for the constructive feedback.
>
> 1. Re: "I see an issue in the high level of arbitrariness in the definition of modularity given... All these choices will give rise to different modules and I see no way in which one should prefer one scheme with respect to the other."
>
> As we elaborated in our general response, there are two independent components: (1) partitioning generation and (2) modularity evaluation. For (1), we do not consider the partitioning generation algorithm as part of our operationalization of modularity. For component (2) we assume that a partitioning is given (using whatever algorithm), and then we can assess whether the network is modular under this partitioning. In other words, our definition of modularity depends on partitioning and it is based on the importance and coherence, which we find a natural proxies. We would like to ask the reviewer whether they consider these two measures as arbitrary?
>
> Finally, to conduct an empirical investigation we needed a way to generate neuron-level partitioning, so we adopted the global weight-based method of Filan et al. (2021) to create multiple variants of its partitioning technique (global vs. local, weight-based vs. activation-based).
>
> 2. Re: "I think that all the heuristic choices described in point 2 make the method difficult to reproduce and to extend, and hence there is a risk for this work to become an isolated publication generating little attention and very few further investigations.""
>
> In fact, our modularity evaluation component can be then applied to *any* neural-level partitioning.
>
> 3. Re: “It is not immediately obvious that the 5 steps described above really provide a good indication of modularity. As a counter example, imagine to train a network with most nodes masked to zero...What is the authors’ opinion on this?”
>
> We'd argue that this network exhibits modularity, albeit a trivial one. With respect to the whole network, the non-mask subnetwork *is* a module - it is a structural part with defined functionality, in this case, using our terminology, it would be at least important.
> Nonetheless, we acknowledge that such configuration might be less of an interest (and one could still apply our method on the non-zeros sub-network for more interesting results), but this is why we consider importance and coherence as proxies to modularity and not a comprehensive definition.
>
> 4. What is the value in knowing that some 'modules' exist if we can never discover what these modules do in practice?
>
> Once subclusters are identified, one could use any interpretability methods they wish to gain an understanding of what they do in practice. But this is not our goal. Instead, we develop quantitative aggregate measures. This is one of the things that makes our paper unique compared to related work.

---

### Official Review · Reviewer_3KTK · 2021-11-02

**Correctness:** 2
**Technical Novelty And Significance:** 2
**Empirical Novelty And Significance:** 2
**Recommendation:** 5
**Confidence:** 4

**Main Review:**

The paper is well written, and generally fair in reporting its findings without much exaggeration. The method proposed here is straightforward (perhaps naïve) on one hand, using rather standard tools for building a neuron similarity graph and clustering it, but on the other hand simplicity is not necessarily a bad thing. The results are intriguing and raise interesting questions, which the authors properly discuss and propose some future work for in Section 5. It is clear that the neuron subclusters here do isolate some encoded information in a way that differs from random clustering, and further it seems this property is robust to the design choices in the proposed procedure.

However, I am not convinced this method really exposes any modularity in neural networks, which is the weakest point of this work and in some sense it does not fulfil its declared goal. The authors do acknowledge some limitations already in their discussion - stating there that it does not provide support for modular design. However, while they claim these results "measure modularity", it is unclear to me whether this is the case, or are they merely measuring redundancy in the network. Indeed, it is typically desired to have multiple pathways in the network encode similar information, as encouraged by dropout and leveraged in network pruning or sparsification (for example, based on the lottery ticket hypothesis). The authors show here that via their partitioning approach, they are able to isolate, and in the lesion case remove, all the neurons that relate to a certain aspect. When compared to random partitioning, the redundancy of information is clearly seen, as random partitions remains more robust or diverse, probably due to redundant information being encoded across many partitions. It is not clear how to go from this evidence of redundancy to saying the network is modular, which would imply that certain subnetworks can serve as individual "modules" that can operate in isolation (to some extent). To establish such evidence, one would at least expect to see some difference between the correlation-based construction (which clearly groups neurons together based on encoding similar information - i.e., providing redundancy) and the connectivity-based construction, which one would hope would provide some submodules of the network. The results on the differences between these approaches (or the global versus local methods) are inconclusive, and in any case there is no attempt here to extract clear modules from the network.

**Summary Of The Paper:**

This work aims to quantitatively evaluate modularity in neural networks (although, as detailed below, I am not sure this is what it ends up doing). To this end, the authors propose to cluster the neurons of the network using spectral clustering applied to a graph that is weighted by similarity between the neurons. They consider two approaches for defining similarity - either via similar responses to inputs (using Spearman or Pearson correlation) or by taking into account the weighted connections (leveraging the learned network weights) between them. There are some technical details regarding the division to layers, exact implementation, and especially the consideration of convolutional architectures (where certain neurons are grouped together along channels), which I will not refer to much here for brevity, but suffices to say that these are all well described in the manuscript and make sense without much need for justification. Once the clusters (or "subclusters" when divided to layers) are obtained, they are evaluated empirically to related them to the network's task. This portion is mostly based on image classification tasks, so clusters are evaluated based on their impact on classification accuracy, their importance in identifying each class, or coherence of visual features captured by them. The results are somewhat inconclusive (more details below), but it is appreciable that the authors do acknowledge this in a well balanced discussion section.

**Summary Of The Review:**

The approach provided in this work is nice (albeit somewhat simple or even straightforward) and provides a way to cluster together neurons that have some redundancy in their captured information, and therefore distil important information pathways or features in the network. The results are intriguing, but in my opinion do not lead to a clear conclusion, and I don't see how to leverage them towards better understanding of neural networks, their properties, or their design. As such, the impact and significance of this work are rather minor. Most importantly, while the paper claims to establish some quantitative approach to measure modularity in neural networks, the results mostly establish the redundancy of information without relating it in a clear way to modularity per se. Therefore, I believe this work is unfortunately slightly below the bar for acceptance to ICLR.

---

> ### Author Response · Authors · 2021-11-17
> **Response to 3KTK**
>
> Thank you for the constructive feedback.
>
> Re: “However, while they claim these results "measure modularity", it is unclear to me whether this is the case, or are they merely measuring redundancy in the network… It is not clear how to go from this evidence of redundancy to saying the network is modular, which would imply that certain subnetworks can serve as individual "modules" that can operate in isolation (to some extent).”
>
> We have two thoughts related to this.
>
> 1. Redundancy meets our definition of modularity.
>
> Redundancy among neurons falls under the definition of modularity we offer in the first sentence of the abstract and in the intro: “A neural network is modular to the extent that parts of its computational graph (i.e. structure) can be represented as performing some comprehensible subtask relevant to the overall task (i.e. functionality).” If you do not agree with this operationalization, could you elaborate on why and recommend what change you would make?
>
> On this note, we consider only the evaluation component (two proxies) as our core operationalization of modularity (see the second point in the general response). Indeed, our proxies would capture redundancy as a form of modularity *if* the given partitioning puts the redundant neurons in separate subclusters.
>
> 2. To the extent that one is interested in forms of modularity that do not relate to redundancy, they still have reason to be interested in our work. For two reasons we think that redundancy is not the whole story in our experiments.
>
> The first reason: as you mentioned, the connectivity-based clusterings are not especially geared to pick up on redundancy, and are in part designed to ensure independence of the resulting clusters. Figure 3 and Table 1 show that these connectivity-based clusterings often score higher on our modularity metrics than the correlation-based clusterings, indicating that the metrics are not only picking up on redundancy.
>
> The second reason: in addition to the experiments in the paper, we also analyzed VGGs trained on CIFAR-10 without regularization. The VGGs analyzed in the paper were trained with dropout and L2 regularization, which we expect to increase redundancy relative to unregularized networks: dropout by encouraging the same information to be encoded in multiple neurons, to be robust to dropping out; and L2 regularization by encouraging weights to be spread out over multiple neurons rather than concentrating on a few. We present the results below. Our gloss is that the unregularized, less redundant networks do not consistently score worse on our modularity metrics than the regularized, more redundant networks, and in fact perform better on importance. We think that comparing results between the regularized and unregularized CNNs highlight that our methods are able to pick up on other forms of modularity besides redundancy.
>
> Importance:
>
> Net | partitioning | p value | effect measure
>
> reg | W/G | *4.40E-8* | 0.837 +- 0.048
>
> reg | W/L | *0.021*     | 0.808 +- 0.041
>
> reg | A/G | *1.62E-11* | 0.926 +- 0.032
>
> reg | A/L | *2.26E-23* | 0.956 +- 0.030
>
> unreg | W/G | *4.99E-12* | 1.041 +- 0.054
>
> unreg | W/L | 0.513          | 0.832 +- 0.033
>
> unreg | A/G | *7.42E-37* | 1.017 +- 0.032
>
> unreg | A/L | *3.26E-48*  | *1.093 +- 0.028*
>
> -----
>
> Coherence (class range):
>
> Net | partitioning | p value | effect measure
>
> reg | W/G | *2.44E-29* | 0.407 +- 0.042
>
> reg | W/L | *2.24E-4*    | 0.692 +- 0.033
>
> reg | A/G | 0.845          | 0.679 +- 0.023
>
> reg | A/L | 0.266           | 0.695 +- 0.021
>
> unreg | W/G | 0.292   | 0.678 +- 0.038
>
> unreg | W/L | *0.007* | 0.873 +- 0.028
>
> unreg | A/G | 0.536    | 0.700 +- 0.025
>
> unreg | A/L | 0.996     | 0.711 +- 0.021
>
> -----
>
> Coherence (vis score):
>
> Net | partitioning | p value | effect measure
>
> reg | W/G | *6.98E-8*  | 0.871 +- 0.011
>
> reg | W/L | 0.442          | *1.013 +- 0.006*
>
> reg | A/G | *5.59E-64*  | *1.327 +- 0.005*
>
> reg | A/L | *7.79E-121* | *1.379 +- 0.004*
>
> unreg | W/G | *0.001*     | *1.034 +- 0.007*
>
> unreg | W/L | 0.744         | 0.993 +- 0.004
>
> unreg | A/G | *2.32E-10* | *1.033 +- 0.003*
>
> unreg | A/L | *9.82E-20*  | *1.050 +- 0.003*
>
> -----
>
> Coherence (softmax H)
>
> Net | partitioning | p value | effect measure
>
> reg | W/G | 0.613  | 1.124 +- 0.012
>
> reg | W/L | 0.616   | 0.992 +- 0.012
>
> reg | A/G | 0.045   | *0.950 +- 0.009*
>
> reg | A/L | *0.013* | *0.930 +- 0.008*
>
> unreg | W/G | 0.202       | 0.994 +- 0.015
>
> unreg | W/L | 0.801       | 1.042 +- 0.011
>
> unreg | A/G | *1.66E-5* | 1.022 +- 0.011
>
> unreg | A/L | *6.00E-4*  | 1.077 +- 0.010

---

### Official Review · Reviewer_d1u7 · 2021-11-04

**Correctness:** 3
**Technical Novelty And Significance:** 2
**Empirical Novelty And Significance:** 2
**Recommendation:** 5
**Confidence:** 3

**Main Review:**

Strengths:
1.	The choice of metrics for modularity seems well-motivated and proposal to quantify the two proxies is seemingly a great strategy.
2.	The paper is very well-written and all the analyses and experiments along with different choices seem reasonable and interesting.
3.	The question of detecting modularity is a very interesting one


Critiques:
1. The paper claims they aim to detect/quantify modularity in DNNs but the proposed technique only seems to address whether spectral clustering-based partitioning can identify modular subsets (to the extent they are `more modular’ than random groups of neurons). The broader question of whether a DNN is modular or not or how its modularity compares against other networks remains untouched in this paper.

2. The results seem more about clusterability than modularity. For instance, an important conclusion I draw from the paper is that the clustering scheme is useful to identify subsets of neurons that jointly serve as a coherent and important subset for a particular task. The extent of modularity quantified by the partitioning is only compared against random groups of neurons. This seems like a very crude baseline and it would have been nice to know if the metrics can be used to compare modularity across networks.

3. Modularity is defined on top of clusterability – but what if the partitioning itself is imperfect. Clusters define candidate modules but what additional value does functional modularity analyses provide in this setting?

4. In the feature visualization experiment, the value of the maximization objective (the ‘score’) might be highly sensitive to the gradient optimization technique – are the authors doing unconstrained optimization in the input space? Importantly, the features highlighted by the visualization technique do not reveal anything interpretable and the visualizations for the true subclusters and random groups of neurons look very similar in all the datasets the authors tested (MNIST, CIFAR10 or ImageNet). Given this, the merits of using partitioning to analyze/interpret DNNs remains unclear.



**Summary Of The Paper:**

This paper seeks to answer whether modern DNNs are modular and proposes statistical methods to quantify modularity. In this study, DNNs are described as modular to the extent they agree with 2 prototypical conditions of modularity, namely importance (which is mainly captured by the drop in accuracy after removing a candidate module) and coherence (which would measure the extent to which neurons in a single module are .activated by similar features). These proxies for modularity, importance (with respect to task performance) and coherence (degree of specialization), are computed based on two well-known DNN interpretability techniques, namely lesions and feature visualization.

**Summary Of The Review:**

While I enjoyed reading the paper and liked how the authors operationalized modularity, I don't think the paper is ready for publication yet as the results and experiment don't seem to add value or strongly support the practical utility of the proposed metrics and means to quantify them in studying DNNs.

---

> ### Author Response · Authors · 2021-11-16
> **Response to d1u7**
>
>
> Thank you for the constructive feedback.
>
> 1. Re: “The broader question of whether a DNN is modular or not or how its modularity compares against other networks remains untouched in this paper.” and “Modularity is defined on top of clusterability – but what if the partitioning itself is imperfect?”
>
> Our sense is that a DNN is modular if there’s a way of dividing up the neurons that satisfies the criteria you want from modularity. In this paper, we develop modularity criteria and apply them to a partitioning method that has already been somewhat explored in the literature. So, our way of addressing the broader question of whether a DNN is modular or not is to take some partitioning methods, develop modularity metrics that measure quantities of interest, and show that DNNs partitioned in these ways score well on these modularity metrics - and are therefore modular.
>
> We feel that comparisons of our metrics between different networks is somewhat informative in allowing us to know which networks have structure that contrasts more with the ‘background noise’ of random groups of neurons in that network, but indeed there is some subtlety here. That said, we felt comfortable not making this our focus, and instead focused on evaluating the modularity of particular partitionings.
>
> 2. Re: “The extent of modularity quantified by the partitioning is only compared against random groups of neurons. This seems like a very crude baseline”
>
> We think that comparisons to random groups of neurons are a good baseline to test whether a clustering exhibits functional modularity within a single network, by testing whether our metrics are higher than they would be for the ‘noise’ of random groups of neurons. Using a less naive baseline would test whether the partitionings we evaluate exhibit modularity in a relative sense rather than an absolute one.
>
> 3. Re: “Clusters define candidate modules but what additional value does functional modularity analyses provide in this setting [of imperfect partitioning]?”
>
> We think that functional modularity analyses reveal the importance and coherence of the candidate modules. This informs future analysis: if a candidate module weren’t important, then it would not be valuable to study its function, and if a candidate module weren’t coherent, then it would not be clear that it has one single function.
>
> Regarding the possibility of imperfect partitioning, we make no claims to identify the optimal modularity structure within networks: just that some structure is there that is to some degree amenable to analysis.
>
> 4. Re: “In the feature visualization experiment, the value of the maximization objective (the ‘score’) might be highly sensitive to the gradient optimization technique – are the authors doing unconstrained optimization in the input space”
>
> We attempt to use standard methods. As detailed in A.5:
> > For small MLPs, small CNNs, and mid-sized CNNs, we generated images using random jittering and scaling, and for ImageNet models, we used Lucid’s default transformations which consist of padding, jittering, rotation, and scaling with default hyperparameters. For all networks, we used the standard pixel-based parameterization of the image and no regularization on the Adam optimizer for 100 steps.
>
> 5. Re: “Importantly, the features highlighted by the visualization technique do not reveal anything interpretable and the visualizations for the true subclusters and random groups of neurons look very similar in all the datasets the authors tested.. Given this, the merits of using partitioning to analyze/interpret DNNs remains unclear.”
>
> One point is that for more coherent subclusters, features highlighted by the visualization technique better summarize the function of those subclusters than the visualizations of random groups of neurons. Indeed, this is a fact which would not be obvious without the measurements we develop in this paper.
>
> That said, it’s indeed true that these visualizations don’t make it obvious what the subcluster is representing. To get a better sense of this, further interpretability techniques would need to be used. What we believe is that more coherence makes a subcluster a better candidate for analyzing as a group, and that these hints are likely to facilitate the analysis of modules.
>
> 6. Re: “the results and experiment don't seem to add value or strongly support the practical utility of the proposed metrics and means to quantify them in studying DNNs.”
>
> It is our hope that research like this that illustrates and helps to analyze structure in DNNs will eventually be practically useful, even though we have not made practical use of them yet.

---

### Author Response · Authors · 2021-11-16
**Global Response from Authors**

In addition to each reviwer’s response, we would like to comment on two matters that more than one reviewer raised.

1. Why we do not focus on developing interpretations of individual clusters.

Our goal in this paper is to quantitatively assess how well a partitioning on a network exhibits modularity. It is not our goal to ascribe interpretations to individual subclusters. Doing so would give us information about individual subclusters, but not about the presence of modularity in the network overall. We think the larger issue of quantifying modularity overall is more novel and needed in the literature. We emphasize that while our paper has strong relations with interpretability work, it is not itself an interpretability paper. Our work is more focused on developing a basic understanding of networks than on interpreting them.

2. How to partition a network and how to evaluate a partition for modularity are separate matters--our main focus is on the latter:

There are two steps involved in evaluating how well a partitioning captures modularity. First, we need a partitioning and second we need to evaluate it for modularity. We are focused on the second part, but this requires that we use *some* partitioning method. In our case, we use ones that involve spectral-clustering because it is well-fit to the problem, thoroughly studied, scalable to ImageNet networks, and previously used by Filan et. al 2021 in deep networks. We could test partitioning methods other than the four that we use, but partitioning for us is a means to an end, and we stick to the four methods that we use because (a) we find they are sufficient for detecting modularity, and (b) we think that methods for detecting modularity given a partitioning are more interesting and useful to current research than comparing partitioning methods.

---

### Decision · Program_Chairs · 2022-01-20

**Decision:**

Reject

**Comment:**

Paper studies if DNNs are modular and proposes statistical methods to quantify modularity.
cluster the neurons of the network using spectral clustering applied to a graph that is weighted by similarity between the neurons.

While the reviewers find the question of modularity relevant, they raise the issue that the results are inconclusive regarding the main stated contribution of the paper (i.e., if modularity is appropriately measured). After discussion, some concerns are answered. However, the main problem of inconclusive results stands. Therefore, this borderline paper is rejected.